# Combination of Water-Saving Irrigation and Nitrogen Fertilization Regulates Greenhouse Gas Emissions and Increases Rice Yields in High-Cold Regions, Northeast China

**DOI:** 10.3390/ijerph192416506

**Published:** 2022-12-08

**Authors:** Yu Sun, Yongcai Lai, Qi Wang, Qiulai Song, Liang Jin, Xiannan Zeng, Yanjiang Feng, Xinrui Lu

**Affiliations:** 1Institute of Crop Cultivation and Tillage, Heilongjiang Academy of Agricultural Sciences, Harbin 150086, China; 2Key Laboratory of Germplasm Enhancement and Physiology and Ecology of Food Crop in Cold Region, Ministry of Education, Harbin 150030, China; 3Heilongjiang Academy of Agricultural Sciences, Harbin 150086, China; 4Plant Nutrition and Resources Institute, Beijing Academy of Agriculture and Forestry Sciences, Beijing 100097, China; 5Northeast Institute of Geography and Agroecology, Chinese Academy of Sciences, Changchun 130012, China

**Keywords:** intermittent irrigation, greenhouse gases emissions, methane, nitrous oxide, rice field, high-cold

## Abstract

Increased rice production, which benefitted from cropping areas expansion and continuous N applications, resulted in severe increases in greenhouse gases (GHG) emissions from 1983 to 2019 in Heilongjiang Province, China. Therefore, field trials were performed in the high-cold Harbin region, Northeast China, to determine the efficiency of incorporating water regimes with N fertilization in minimizing the impact of rice production on GHG emissions. Two water-saving irrigation strategies, intermittent irrigation (W1) and control irrigation (W2), were used relative to continuous flooding (W0), and we combined them with six fertilized treatments. Our results demonstrated that W1 and W2 significantly decreased seasonal CH_4_ emissions by 19.7–30.0% and 11.4–29.9%, enhanced seasonal N_2_O emissions by 77.0–127.0% and 16.2–42.4%, and increased significantly yields by 5.9–12.7% and 0–4.7%, respectively, compared with W0. Although trade-offs occurred between CH_4_ and N_2_O emissions, W1 and W2 resulted in significant reductions in global warming potential (GWP). Moreover, low N rates (<120 kg N ha^−1^) performed better in GWP than high N rates. N fertilization and irrigation regimes had remarkable effects on rice yields and GWP. In conclusion, the incorporation of W1 and a N application under 120 kg N ha^−1^ could simultaneously mitigate GWP while enhancing production in black soils in high-cold Northeast China.

## 1. Introduction

Rice (*Oryza sativa* L.) is an important food crop for global food security, feeding roughly half of the world’s population; furthermore, its demand is expected to increase by 28% in 2050 [1]. However, rice farming has been identified as a major anthropogenic source of greenhouse gas (GHG), contributing up to 11% and 30% of global anthropogenic emissions of methane (CH_4_) and nitrous oxide (N_2_O), respectively [2,3], which is approximately four times higher than that of either wheat or maize farming [4]. Therefore, identifying effective management techniques for agroecosystems is crucial to reduce the impact of rice production on global warming.

Irrigation methods and N fertilization applications are the two main drivers for rice production, not only in optimizing rice yield but also regulating N_2_O and CH_4_ emissions [5,6]. Mitigation strategies for the impact of rice production on global warming should focus on modifications to current water and N management practices. Intermittent irrigation, as a “win–win” management strategy, has become increasingly popular, not only in inhibiting ineffective tillers, removing toxic substances, and improving root activities, but also reducing water consumption and enhancing water-use efficiency compared with continuous flooding [7,8,9] in China and many other countries [10,11]. Furthermore, intermittent irrigation, defined as a period of drainage lasting several days in the middle of the rice season, and drying–wetting alternation during the growing season, alter anaerobic and aerobic cycling, thereby affecting various processes underlying CH_4_ and N_2_O production. It has gained increasing attention as a promising strategy for mitigating GHG emissions in rice agroecosystems.

Numerous studies have shown that the intermittent irrigation of paddy fields reduced seasonal CH_4_ emissions [12,13,14] while simultaneously stimulating N_2_O emissions, resulting in a trade-off between CH_4_ and N_2_O emissions. Although CH_4_ and N_2_O emissions showed a trade-off relationship, switching from continuous flooding to intermittent irrigation methods considerably reduced the total global warming potential (GWP) of rice production [9,11,13], mainly owing to the reduction in CH_4_ emissions. On the contrary, Jiang et al. [15] reported that non-continuous flooding practices reduced CH_4_ emissions by 53% and increased N_2_O emissions by 105%, which contributed 12% on average to the combined global warming potential compared with continuous flooding by conducting a meta-analysis using 201 paired observations from 52 studies. More importantly, the magnitude of trade-offs between CH_4_ and N_2_O emissions from intermittent irrigation varied with soil type, drainage timing, drainage interval, duration, soil-drying severity, water stress level, and other management factors, such as fertilization [9,13,16]. Evidence suggested that CH_4_ and N_2_O emissions from rice fields can be strongly affected by N fertilization management [16,17,18,19].

N_2_O is primarily produced by nitrification and denitrification processes and influenced by available soil substrates (NO_3_^−^, NH_4_^+^) [20,21]. Previous studies have sufficiently documented that N_2_O emissions from rice systems increased as N fertilization rates increased [6,22,23]. In contrast, when N fertilization inputs exceeded rice demand, a threshold response of N_2_O emissions to N fertilization rates was met [24]. The contributions of exogenous N fertilization for N_2_O production in rice fields using intermittent irrigation techniques are not well understood. Chemical fertilizations affect CH_4_ emissions from rice fields mainly through the following three processes: the production, transport, and oxidation of CH_4_ emissions [25]. Conflicting results regarding the effects of N rates on CH_4_ emissions during the rice-growing season have been reported, including stimulation [23,26], no effects [5], or inhibition [17], which are dependent on N fertilization species and rates [5,27,28]. Thus, an optimal fertilization regime is critical not only for regulating CH_4_ and N_2_O emissions but also for sustaining crop yields during rice growing.

Heilongjiang Province is a major rice-growing region where the black soil belt runs north to south, producing 27 million rice grains (4.0% of the total grain production in China) in 2019 [29]. The black soil zone in cold Northeastern China is the most important commodity grain base in China, and it is hailed as the “stabilizer” and “ballast stone” of China’s grain production due to its inherently fertile and productive characteristics. The rice-growing area markedly increased from 0.24 to 3.81 million ha between 1983 and 2019, and the total chemical fertilization application increased from 10.62 × 10^5^ t in 1983 to 52.93 × 10^5^ t in 2019 [29]. The expansion of rice areas and increases in N application levels not only resulted in significant GHG emissions [30,31] but also significant irrigation water consumption, causing the groundwater table to decline, which increased pumping costs for farmers. The importance of interactive effects between intermittent irrigation and N fertilization has been widely recognized, with the majority of research focusing mainly on double-cropping paddy fields in Southern China [5,16,26,31,32]; however, the interactive impacts still remain elusive. For instance, more recently, Shi et al. [16] investigated the effects of reduced irrigation and N application on GHG emissions in double-cropping rice paddy fields in Jiangxi Province, China. Their results indicated that early- and late-season rice decreased CH_4_ emissions by 14.5–37.4% and 16.7–52.3%, respectively, whereas they increased N_2_O emissions by 12.5–35.3% and 23.1–34.1%, respectively, under the same N fertilization management processes. This interaction involved a complex system that regulates GHG dynamics and yields variable cumulative emissions across soil types, rice-growing seasons, and locations, with large spatio-temporal variations [33]. The black soil zone in Northeastern China experiences long and cold winters and possesses relatively high levels of organic content in the soil. Thus, only single-cropping rice can be grown, as opposed to the major double-rice producing areas in Southern China. Therefore, the urgent need to mitigate GHG emissions while maintaining high rice yields has spurred research into the optimal interrelation between intermittent irrigation and N fertilization in cold black soil. However, not much is known about the measurement of GHG emissions at higher latitudes in cold regions, particularly the interactive impacts of water management and N fertilization on CH_4_ and N_2_O emissions. Thus, this study hypothesized that increasing N fertilization would reduce both N_2_O and CH_4_ emissions, and that water-saving irrigations would reduce CH_4_ while increasing N_2_O emissions beneficial to decreasing the GWP. Therefore, we conducted a field experiment at higher latitudes in cold Heilongjiang, Northeast China, with the objectives of (i) testing the individual positive or negative effects of N fertilization and water-saving irrigations on CH_4_ and N_2_O emissions, as well as its specific contributions to the GWP; (ii) evaluating the combined impacts of water-saving irrigations and N fertilizations on the GWP as well as rice production; and (iii) identifying efficient fertilization and water management practices to optimize rice yields while minimizing GHG emissions. The outcomes could make significant contributions toward achieving “Peak Carbon Dioxide Emissions” as well as sustainable rice crop production under a limited water resource regime.

## 2. Materials and Methods

### 2.1. Study Area

We conducted the field experiment in 2018 at the National Modern Agricultural Demonstration Park’s experimental station in Minzhu Town, Harbin City, China (45°49′ N, 126°48′ E.) on chernozem soil (Mollisols in USA-ST) (Figure 1). The site has an elevation of 117 m above mean sea level. The climate is cold temperate continental monsoon, with a mean annual temperature of 4.5 °C, and has mean winter and summer air temperatures of −18 °C, and 24 °C, respectively, as well as an effective accumulated temperature (≥10 °C) of 2968 °C. It has an annual sunshine time of 2668 h, a frostless period of 131–146 days, and a mean annual precipitation of 585 mm in the last 20 years, characterized by one main rainfall period in summer (July–September). This region is dominated by a typical single-cropping rice system. The soil type is Phaeozem, and the basic properties of soil were as follows: pH (1:5 soil:water) 7.4, total N 3.45 g kg^−1^, total P 0.89 g kg^−1^, total K 31.69 g kg^−1^, available N 79.77 mg kg^−1^, available P 56.81 mg kg^−1^, available K 167.41 mg kg^−1^, and soil organic matter 29.69 g kg^−1^.

### 2.2. Crop Management and Treatments

We conducted a split-plot experiment involving the irrigation regime and N rate in this study. The irrigation regime was the main factor comprising: (1) continuous flooding (W0), continuously flooded with a water depth of 10–15 cm until 1 week before crop harvesting; (2) intermittent irrigation (W1), submerged in a 0–2 cm water depth from transplanting to re-greening and then applied to a 3–7 cm water depth at a 7–9-day interval. This process was repeated until milk stage and then drained one week before rice harvest; and (3) control irrigation (W2), submerged in a 0–2 cm water depth from transplanting to re-greening and then applied to a 20–30 cm depth of water and finally drained one week before rice harvest. The second factor was N fertilization, which had six levels: no fertilization (N0, control), 60 kg N ha^−1^ (N60), 80 kg N ha^−1^ (N80), 100 kg N ha^−1^ (N100, 120 kg N ha^−1^ (N120), and 140 kg N ha^−1^ (N140). There were 18 treatments in total, and each treatment had three replicates. Each plot was 4 m × 4 m and was isolated by brick concrete, 1.2 m deep, between any two adjacent plots to prevent water and fertilization exchange. We set all plots under a rain shelter to control irrigation water usage.

We sowed a major japonica rice variety in this region, namely “Longdao 21”, in a nursery bed on 14 April; then, we transplanted rice seedlings to the paddies on 16 May and harvested on 19 September 2018. We applied the total amount of urea as N fertilization in three interventions: 50% as a basal application on 16 April, 30% as a re-greening topdressing on 25 May, and 20% as a tillering topdressing on 12 June. We employed P fertilization (P_2_O_5_, 55 kg ha^−1^) as the basal fertilization in all treatments. We applied potassium fertilization (K_2_O, 50 kg ha^−1^) with 50% as a basal application on 16 April and 50% at the returning green stage on 25 May.

### 2.3. N_2_O and CH_4_ Measurements

We simultaneously collected GHG samples using the static chamber technique. Transparent chambers with dimensions 40 cm × 30 cm × 50 cm (length × width × height) were made of acrylic sheets. Before rice transplanting, we permanently installed a PVC soil base frame (channel) of 15 cm height and 5 cm internal diameter in each plot to ensure reproducible placement of the gas-collecting chamber during the rice-growing period. The top edge of the frame had a groove (5 cm in depth and 2 cm in width) filled with water to ensure that the system was airtight during gas sampling. We attached a small battery-operated rotary fan to each chamber to guarantee complete gas mixing. Each chamber was wrapped with a layer of sponge and aluminum foil to minimize the air temperature changes inside the chamber during gas sampling. We collected gas samples from a silicone septum on top of the chamber by inserting a 50 mL airtight syringe attached to 24-gauge hypodermic needles with a 3-way stopcock into pre-evacuated vacuum tubes at 0-, 10- and 20-min intervals after closure. We recorded the headspace volume inside the chambers to calculate N_2_O and CH_4_ concentrations. We chose gas samples to correspond as closely as possible to the key growth stages of rice, which were collected from 9:00 a.m. to 11:00 a.m. on sunny days. We used an attached digital thermometer to measure the temperature inside the chamber. We stored gas samples in syringes and then in evacuated vials for <1 day before analysis on a gas chromatograph. We analyzed the gas samples using a modified gas chromatograph (Agilent 7890A, UMPA, Palmyra, PA, USA) equipped with a flame ionization detector (GC 8 A Series, Shimadzu, Japan) and an electron capture detector to obtain the CH_4_ and N_2_O concentrations. We equipped the gas chromatograph with a flame ionization detector (FID) to detect CH_4_ and an electron capture detector (ECD) to detect N_2_O. We used nitrogen and a gas mixture of argon and methane (95% Ar-5% CH_4_) as the carrier gases for CH_4_ and N_2_O, respectively. The oven temperature was 55 °C and the FID temperature was 200 °C. We determined fluxes of CH_4_ and N_2_O from the slope of the mixing ratio change with the four sequential samples. We only selected available sample sets when they followed a linear correlation coefficient value of R^2^ greater than 0.90. We sequentially accumulated cumulative CH_4_ and N_2_O emissions for the entire cropping period from the emissions averaged on every two adjacent measurement intervals [31,34]. We calculated the fluxes of CH_4_ and N_2_*O* using linear regression of the measured concentrations against sampling time.
(1)$$F=\rho h\frac{273}{273+T}\;\frac{\mathrm{dc}}{\mathrm{dt}}$$
where F is the gas flux rate (mg m^−2^ h^−^^1^), ρ is the gas density in the standard state, h is the height of the chamber above the water surface (m), dc/dt is the slope of the linear regression for gas concentration gradient through time, and T is the mean air temperature inside the chamber during sampling (°C). We calculated the cumulative gas emissions during the study period by integrating the gas emissions between every two adjacent measurement intervals [9].

We calculated the GWP of CH_4_ and N_2_O at 100-year time horizon using the following equation [35]:(2)$$\mathrm{GWP}=\left(t\;{\mathrm{CO}}_{2}{\;\mathrm{equivalent}\;\mathrm{ha}}^{-1}\right)=298\times {\mathrm{GWP}}_{\left(N_{2}O\right)}+34\times {\mathrm{GWP}}_{\left({\mathrm{CH}}_{4}\right)}$$
where GHGI represents greenhouse gas intensity.
(3)$$\mathrm{GHGI}(t\;{\mathrm{CO}}_{2}{\;\mathrm{equivalent}\;t}^{-1})=\frac{\mathrm{GWP}}{\mathrm{rice}\;\mathrm{yields}}\;\;\;\;\;$$

### 2.4. Plant Sampling and Estimation of Yields

At harvesting, we collected rice grains within 2 m × 2 m frames of each plot to calculate yield and yield components in triplicate. We air-dried and adjusted the grains to 14% moisture content. We also calculated the tiller number, seed-setting rate, 1000-kernel weight, and grain numbers per spike for each plot.

### 2.5. Data Analysis and Statistics

We conducted analysis of variance (ANOVA; least significant difference method) to assess the effects of N fertilization and water regimes on N_2_O and CH_4_ emissions, GWP, GHGI, and rice yield among treatments. All data were presented as mean ± SE (n = 3). We performed a pairwise mean comparison of treatments using Tukey’s honest significant difference test at a 5% level of probability (*p* ≤ 0.05). We performed the calculations with SPSS (version 21.0; SPSS Inc., Chicago, IL, USA) and drew the figures using Origin 12 (Origin Lab Corporation, Northampton, MA, USA).

## 3. Results

### 3.1. Climatic Data

Figure 2 depicts the climatic data for the rice-growing season. Throughout the rice-growing season, the mean temperature was 20.1 °C, with the mean maximum threshold recorded in July. On 16 April and 18 June, the maximum and minimum air temperatures were recorded at 1.6 °C and 38.0 °C, respectively. The total rainfall and sunshine times were 543.2 mm and 1581 h, respectively.

### 3.2. CH_4_ and N_2_O Fluxes

The dynamics of the measured CH_4_ fluxes are presented in Figure 3. They were characterized by a unimodal curve, which showed that the CH_4_ emissions steadily ascended with time, peaked at the tillering stage, gradually decreased by the jointing–booting stage, and then remained at low levels until the end of the rice season. Moreover, the CH_4_ emissions peaks reached as high as 40.47 mg m^−2^ h^–1^, 32.95 mg m^–2^ h^–1^, and 36.25 mg m^–2^ h^–1^ on average for the W0, W1, and W2 treatments, respectively. The average CH_4_ flux was highest for W0 (13.44 mg m^−2^ h^−1^), followed by W1 (10.06 mg m^−2^ h^−1^), and then W2 (10.87 mg m^−2^ h^−1^), except for the re-greening and maturation stages at the same N fertilization level. In all the water regimes, CH_4_ emissions were stimulated by low N promotion and prevented by high N fertilization (Figure 3).

Overall, N_2_O emission patterns were relatively low compared with CH_4_ emission patterns during the rice-growing seasons (Figure 4). Application of N fertilization at the tillering stage caused a sharp increase in flux in all treatments, which peaked at the jointing–booting stages. Except for a few small peaks at the maturation stage, the fluxes then gradually declined and remained elevated (Figure 4). Among the six N fertilization levels, the peaks were in the order of N120 > N140 > N100 > N80 > N60 > N0, except for N80 = N60 under W0. Furthermore, the peak N_2_O fluxes differed significantly among different water regimes. Compared with W0 irrigation, W1 and W2 produced peak N_2_O emissions that were on average 99.0% and 44.1% higher than that of the W0 irrigation after the second dressing, respectively. Among the different treatments, the highest N_2_O fluxes were observed under the W1N120 treatment, while the lowest N_2_O fluxes occurred under the W0N0 treatment.

### 3.3. CH_4_ and N_2_O Emissions

The tillering and jointing–booting periods comprise considerable portions of growing-season CH_4_ emissions (Figure 5), accounting for 38.6% and 26.9%, respectively. The opposite trend in CH_4_ emissions was observed with increases in the N fertilization dosage. For example, mean cumulative CH_4_ emissions under W1 decreased by 28.5% and 32.4% at N120 and N140 compared with N60, despite the fact that there were no significant differences (*p* > 0.05) between them. A comparison of cumulative CH_4_ emissions measured with the different irrigation regimes revealed that the emissions in W1 and W2 significantly decreased by an average of 19.7–30.0% and 11.4–29.9% compared with W0, respectively.

On average, N_2_O emissions during the 10-day drainage at the jointing–booting periods accounted for 34.2% of the seasonal total, which was significantly higher than other growth stages (Figure 6). The effects of irrigation methods and N levels on N_2_O cumulative seasonal emissions were similar to those of CH_4_ emissions, as revealed in Figure 7b. The cumulative seasonal N_2_O fluxes responded strongly to N fertilization applications, increasing with N dosage but decreasing once the dosage exceeded 120 kg ha^−1^. For example, cumulative N_2_O emissions from W1N60 and W1N120 increased by almost 22.1%, whereas the increase between W1N120 and W1N140 slowed down to approximately 4.8%. In terms of irrigation treatment, W1 and W2 significantly decreased the cumulative seasonal N_2_O fluxes by 77.0–127.2% and 16.1–42.4%, respectively, compared with that in W0 under the corresponding N fertilization levels. The combination of W1 and N120 emitted the most N_2_O, approximately three times higher than the combination of W0 and N0, which emitted the least.

### 3.4. Grain Yield, GWP and GHGI

The combined irrigation regime and N fertilization application significantly affected grain yields (Table 1). Compared with W0, W1 and W2 both enhanced the mean rice yield by 5.9–12.7% and 0–4.7%, respectively, showing tendencies in the order of W1 > W2 > W0. Relative to N60, higher N fertilization rates enhanced rice yield by 4.9%, 8.6%, 15.8%, and 11.2% for the N80, N100, N120, and N140 treatments, respectively, under the same irrigation regime. However, there were no significant differences (*p* < 0.05) between N120 and N140 (Table 1). W1N120 produced the highest grain yield among all the treatments (Table 1). For the components of rice yield, the tillering number and grain number per spike with N fertilization application followed a similar tendency to yield except for 1000-kernel weight (Table 1). Under the corresponding N fertilization treatment, W1 significantly increased grain number per spike by 23.2–40.9% and 7.7–31.5% than W0 and W2 (*p* < 0.05) (Table 2).

The GWP induced by N_2_O and CH_4_ emissions during the rice-growing seasons is presented in Figure 7c. A comparison of the GWP across all the treatments showed that CH_4_ contributed 97.9–99.5% of GWP during the rice-growing season. While water regimes had a significant effect on the GWP, when comparing the corresponding treatments with W0, W1, and W2, the GWP was reduced by 18.7–29.4% and 11.1–19.5%, respectively. The GWP increased and then decreased with the increasing N dosage (Figure 7c). Within W1, compared with N0, the GWP for N60, N80, and N100 were significantly higher at 29.6%, 15.5%, and 3.6%, respectively (*p* < 0.05), whereas GWP for N120 and N140 were lower at 6.5% and 11.6%, respectively.

Under the same irrigation pattern, GHGI tended to decrease with increasing N dosage (Figure 7d), with the largest reduction occurring at a dosage of 140 kg N ha^−1^ by 22.5% when compared with N0. When the N rate exceeded 120 kg N ha^−1^, the mitigating effects of N applications on GHGI tended to decrease. For example, W1N80, W1N100, W1N120, and W1N140 significantly decreased GHGI by 1.2%, 14.0%, 29.8%, and 30.4%, respectively, relative to N0 (*p* < 0.05), with the exception of W1N60. Similar to the GWP, W1 and W2 significantly reduced the GHGI by 26.5–33.5% and 11.3–31.1% relative to W0. Moreover, W0 exhibited a greater warming effect on GHGI, and in combination with N60, resulted in the highest seasonal GHGI (3.78 ± 0.11 kgCO_2_ equivalent kg^−1^), whereas with N140, it resulted in the lowest seasonal GHGI (1.5 ± 0.1 kg CO_2_ equivalent kg^−1^). The two-way ANOVA confirmed that the irrigation regime and N fertilization significantly affected GHGI, and their interaction had a considerable impact (Table 3; Figure 7d). These results indicated that water-saving irrigation reduced the GWP of rice per unit yield. Under water-saving irrigation, a high N rate (N140) significantly (*p* < 0.05) reduced GHGI compared with N0, N60, N80, and N100; however, no significant difference was observed compared with N120 treatments (*p* > 0.05).

## 4. Discussion

### 4.1. CH_4_ Emissions

This study showed that CH_4_ emissions from rice fields in Heilongjiang Province exhibited a single-peak variation trend during the rice-growing period, with lower emissions at the regreening stage, peaking at the tillering stage, and then gradually decreasing. This is in line with previous research [36]. There are three possible interpretations for this result. First, the variational depth of the floodwater regulated the fluctuations in soil oxidation reduction conditions. Second, the relatively lower temperature in Heilongjiang Province suppressed CH_4_ emissions owing to slow organic matter degradation and low methanogens activity. Finally, the rising temperature enabled rice to grow stronger and more vigorously, increasing soil C availability for CH_4_ production [5]. Water regime influences CH_4_ emissions from rice fields, which can be attributed to several factors (e.g., soil moisture, O_2_ content, and organic matter degradation) that impact CH_4_ production, oxidation, and transportation [37]. Furthermore, CH_4_ fluxes and cumulative emissions significantly differed between the irrigation treatments. The higher CH_4_ emissions that occurred in W0 have been reported in similar studies [9,13,38]; however, our experiment provided further evidence that W0 stimulated production and substantially increased CH_4_ emissions in rice fields because the long-term high soil water content resulted in a reduction in soil redox potential, thus creating a favorable condition for CH_4_ production [9,37]. Similar measurements were also obtained where W1 exhibited the lowest peak and significantly lower cumulative CH_4_ emissions than W2 and W0 (Figure 3) over the entire growing season as a result of soil water conditions created by drying–wetting cycles after the tillering stage. This frequently created an aerobic soil environment favorable for inhibiting methanogenesis while accelerating methanotrophy, thus reducing CH_4_ emissions. This may be because intermittent irrigation promotes root system development, which increases the redox potential in the rhizosphere due to the strong root oxygen secretion ability, which ultimately inhibits CH_4_ production [6]. The net effect of CH_4_ production and oxidation in the soil is CH_4_ emissions from paddy fields. Zhang et al. [12] illustrated the mechanisms by which W1 significantly reduced CH_4_ production potential while slightly increasing CH_4_ oxidation potential with a stable carbon isotope technique. In the present study, CH_4_ fluxes and cumulative emissions were stimulated by low N rates; however, they were prevented by high N rates. Increasing N rates from 60 to 120 kg N ha^−1^ were consistent with the findings of Banger et al. [39], who reported that CH_4_ emissions were stimulated at low N rates but inhibited at high N rates across 33 study sites and 155 data pairs worldwide. A possible explanation is that the higher urea input might decrease soil pH and provide an unfavorable redox potential for methanogen growth and activity. Another possible explanation is that the availability of NH_4_^+^ enhances the activity of methanotrophic bacteria, thereby stimulating CH_4_ consumption. However, this differs slightly from the findings of Dong et al. [6], who showed that high-input N fertilization (150 and 225 N ha^−1^) had no effect on CH_4_ emissions in the same region of Harbin, China. This discrepancy may be attributed to the different floodwater depths and N rates. Dong et al. [6] used a floodwater depth of 5 cm rather than the 1–3 cm in the present study, and the higher floodwater depth caused significant N leaching losses, lowering available N concentrations. On the contrary, some other studies have found that increased N fertilization can promote CH_4_ production [40] and had no significant stimulation or mitigation effect by meta-analyses [22]. The mechanism of the effect of N fertilization on CH_4_ emissions from rice fields is complex and requires further investigation. The interaction between water and N fertilization had a significant effect on seasonal cumulative CH_4_ emissions (*p* < 0.01). In line with our hypothesis, the timing of irrigation altered how CH_4_ emissions changed when N fertilization increased.

### 4.2. N_2_O Emissions

N_2_O emissions are mainly produced by soil microbial nitrification and denitrification processes in the soil, which vary largely with irrigation regimes and the increase in N inputs in rice paddies [6,9]. In our analysis, the seasonal cumulative emissions of N_2_O increased as the N rate increased. Our data support previous studies that reported that the application of inorganic N fertilization enhances the NH_4_-N and NO_3_-N (substrate for nitrification and denitrification) content of soil [17], thus increasing N_2_O emissions. However, when the N rate exceeded 120 kg ha^−1^, the emission decreased, though the difference was not significant (*p* > 0.05), indicating a nonlinear response corresponding with the III model theory of Kim et al. [41]. These results agree with the findings of Pittelkow et al. [24] on the threshold effect of N fertilization rate on N_2_O emission. As N input continues to exceed the capacity of soil microbes to take up and utilize N, the emissions of N_2_O would slow down and eventually reach a steady state [6,41]. Water management significantly affected cumulative N_2_O emissions (Figure 7b). W1 and W2 increased N_2_O emissions by 16.1–42.4% and 77.0–127.2%, respectively, under identical fertilization conditions compared with W0, which is likely a result of the aerobic conditions that facilitated thorough nitrification due to improved aeration in W1 and W2, particularly alternate anaerobic and aerobic cycling in W1 compared with constant anaerobic conditions in W0. This results in increased nitrification and incomplete denitrification due to improving soil redox potential in accordance with the previously reported studies [9,42,43]. Thus, by enhancing the oxygenated soil pores, the cumulative N_2_O emissions of N120 increased by 27.4% compared with N60, whereas W2 irrigation increased by 16.09–42.4% and 77.0–127.2% compared with W0 and W1 irrigation. These findings demonstrated that the irrigation method is a more important factor for N_2_O emissions [40,44] and that N fertilization application has a significantly lower effect on N_2_O emissions than water management. Furthermore, under water-saving irrigation, the seasonal cumulative N_2_O emissions of the high N fertilization treatments were lower in W1 and W2 than that of the low N fertilization treatment, and the interaction of water and N fertilization had a significant effect on N_2_O (*p* < 0.01). This suggests that the effect of N fertilization on N_2_O emissions can only be stimulated under appropriate water conditions; therefore, the interactions between water and N fertilization cannot be ignored. Furthermore, the individual effects of N addition and water amendment on N_2_O fluxes are consistent with our hypothesis.

### 4.3. GWP, Yields and GHGI

In this study, the GWP was used as an indicator for the relative incidence of CH_4_ and N_2_O emissions in overall GHG emissions under different irrigation systems in combination with N fertilization application (Figure 7c). As expected, the GWP was mainly attributed to CH_4_ emissions, covering more than 97.9% during the rice-growing period. These findings were consistent with previous studies [6,43], which showed that the total GWP in rice fields was positively correlated with that of CH_4_ emissions. Despite this, our contribution rate of CH_4_ to GWP was higher than the results of Dong et al. [6] in the same high-cold regions. Based on our measurements, the adoption of W1 and W2 resulted in a mean decrease in the GWP by 26.3% and 21.6%, respectively, relative to W0. These reductions underlined that water-saving irrigation could effectively mitigate GWP, even if the decrease in CH_4_ emissions was partially offset by a slight increase in N_2_O emissions in the high-cold rice fields. Notably, the mitigation effect in our study was less obvious than that reported by Peyron et al. [45], who observed that intermittent irrigation can effectively reduce emissions by 83%. However, their analysis did not involve N application. Instead, their results were the consequences of combined actions, such as sampling frequency [6], rice variety [43], drainage frequency, duration, and the level of water stress [22]. Our results indicated that W1 irrigation had the most significant impact in the rice fields, followed by W2, which was similar to the findings of Dong et al. [6]. Intermittent irrigation could be an effective tool for reducing total GHG emissions without losing rice productivity [30] by controlling the limiting values of soil water potential related to specific stages and ineffective tillers, enhancing root growth and activities [5], and thereby increasing grain yield [46]. Moreover, by achieving a maximum drying index, the intermittent irrigation can maintain rice yield [47]. The effects of water management on rice yield vary with rice varieties and meteorological conditions. In this study, rice yields were also significantly influenced by N treatments and increased with N rates but decreased when the fertilization rate was higher than 120 kg ha^−1^. If crops competed more with soil N than nitrification and denitrification microbes, N_2_O emissions would not only be reduced, but rice production would also be insured [31], which was confirmed by our analysis (Table 3). Therefore, the W1N120 treatment is the best option for water regime–N fertilization incorporation in this area because it may be the reciprocal relationship between water and N fertilization that increases rice yield.

GHGI can be used to assess both GWP and potential economic benefits from crop production. Our analysis showed that W1 and W2 significantly reduced GHGI by 26.5% and 21.6%, respectively, with an increased rice yield compared with W0, which was within the range of previously reported studies [30] in Bangladesh, demonstrating that the adoption of the intermittent irrigation reduced GHGI by 21–43% compared with conventional irrigation regimes. Except for the N120 level, GHGI levels at high N rates (N140) (*p* < 0.05) were significantly lower than at other N rates under the same irrigation process, and the mitigating effects tended to decrease when the N rate was greater than 120 kg N ha^−1^. These results indicated that N fertilization increased rice yield more than GWP, resulting in a large reduction in the GHGI. Therefore, a balance exists between rice yield increase and GWP reduction by adjusting the N application rate, which is consistent with the results of a meta-analysis by Feng et al. [22]. W1 in combination with N120 had the lowest GHGI, which was proposed as a win–win strategy for mitigating the GWP while maintaining grain yields in rice fields. Although W1 increased grain yield in comparison with W0, the GHGI did not decrease, indicating that it may not achieve an optimal yield. Improving yields and reducing CH_4_ and N_2_O emissions in the cold rice field remain a great challenge for the future, and we hope to spur more work in this direction.

## 5. Conclusions

In this study, we found that water-saving irrigation processes suppressed CH_4_ emissions in comparison with W0. Seasonal cumulative CH_4_ emissions from W1 and W2 decreased by 19.7–30.0% and 11.4–29.9%, respectively, which mainly concentrated during the two growth stages of rice tillering and jointing–booting, whereas seasonal N_2_O emissions in W1 and W2 increased by 77.0–127.2% and 16.1–42.4% respectively, peaking at the jointing–booting stage; thus, the emissions mainly concentrated during this growing stage. The CH_4_ emissions from rice fields had a higher contribution to global warming potential (GWP), accounting for 97.9–99.6%, while N_2_O only accounted for a small portion. Moreover, water-saving irrigation was the best solution for GHG mitigation. W1 and W2 significantly decreased the GWP by 18.7–29.4% and 11.1–29.5%, respectively, compared with W0. The combination of water irrigation regimes and N fertilization had significant effects on rice grain yield, seasonal CH_4_ and N_2_O emissions, GWP, and GHGI. Based on our findings, we suggest that intermittent irrigation in combination with 120 kg N ha^−1^ N applications could be a potential option for mitigating the yield-scaled GWP without sacrificing grain yield in the high-cold rice fields of Northeast China. However, the long-term effects of this strategy on yield and greenhouse gases emissions should be further investigated.

## Figures and Tables

**Figure 1 ijerph-19-16506-f001:**
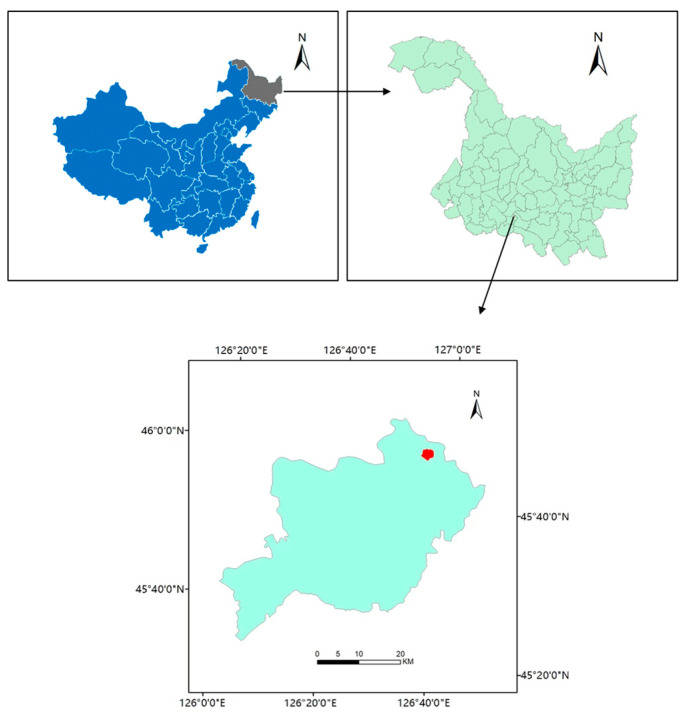
The map of the experimental site.

**Figure 2 ijerph-19-16506-f002:**
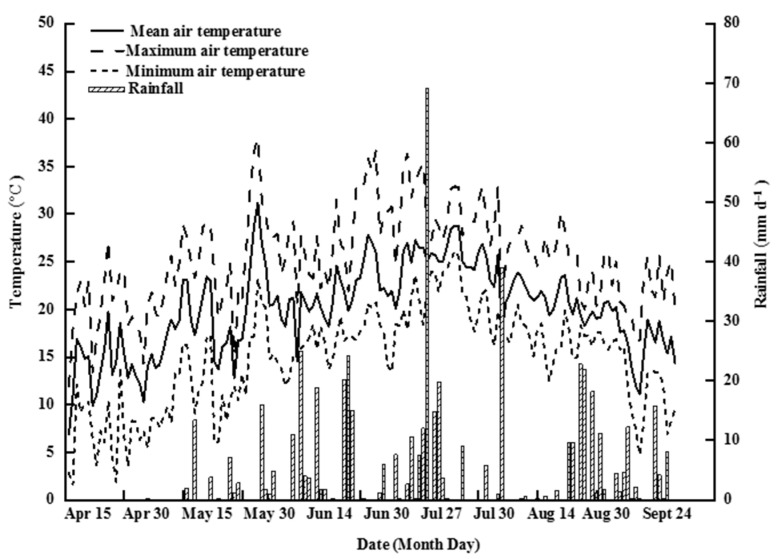
Climatic data for the rice-growing season.

**Figure 3 ijerph-19-16506-f003:**
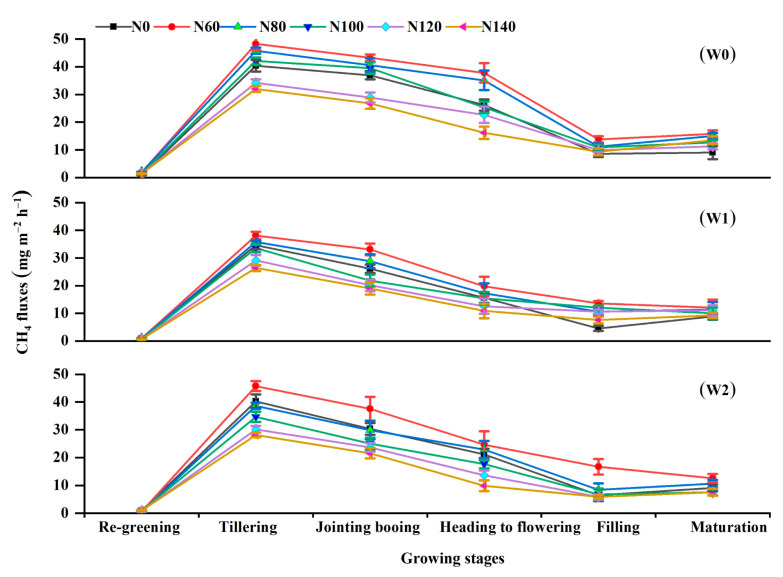
Variations in CH_4_ fluxes as affected by irrigation regimes with different N application rates during the rice-growing period. W0, W1, and W2 represent continuous flooding, intermittent irrigation, and control irrigation, respectively. Urea application at rates of 0 (N0), 60 (N60), 80 (N80), 100 (N100), 120 (N120), and 140 (N140) kg N ha^−1^.

**Figure 4 ijerph-19-16506-f004:**
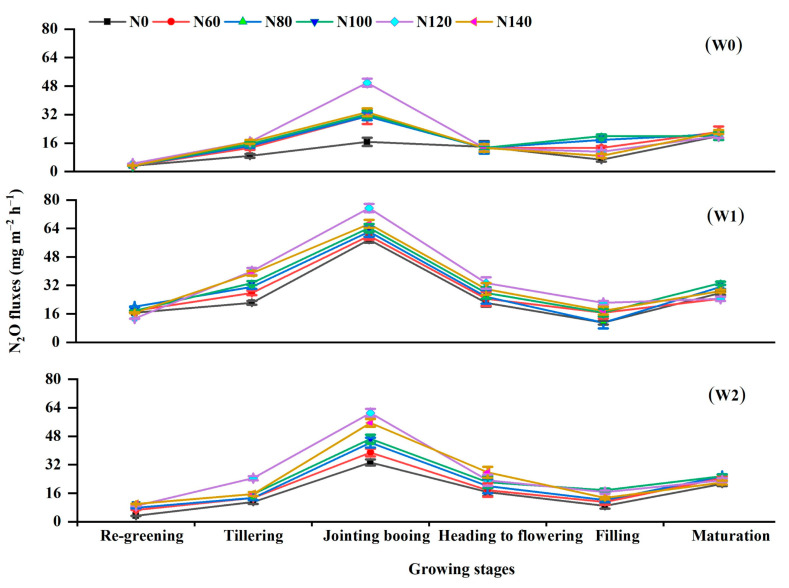
Variations in N_2_O fluxes, as affected by irrigation regimes with different N application rates during the rice-growing period. W0, W1, and W2 represent continuous flooding, intermittent irrigation, and control irrigation (W2), respectively. Urea application at rates of 0 (N0), 60 (N60), 80 (N80), 100 (N100), 120 (N120), and 140 (N140) kg N ha^−^^1^.

**Figure 5 ijerph-19-16506-f005:**
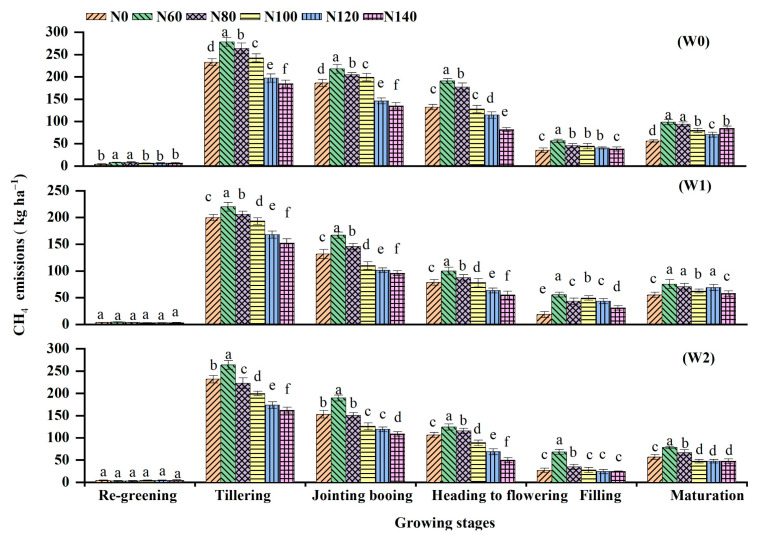
Seasonal CH_4_ emissions affected by irrigation regimes with different N application rates during the rice-growing period. W0, W1, and W2 represents continuous flooding, intermittent irrigation, and control irrigation (W2), respectively. Urea application at rates of 0 (N0), 60 (N60), 80 (N80), 100 (N100), 120 (N120), and 140 (N140) kg N ha^−1^. Values followed by the different letter are significant different at *p <* 0.05.

**Figure 6 ijerph-19-16506-f006:**
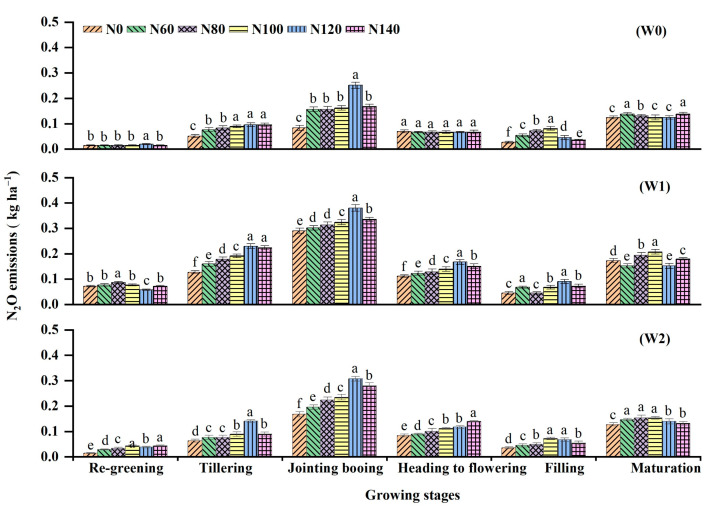
Seasonal N_2_O emissions affected by irrigation regimes with different N application rates during the rice-growing period. W0, W1, and W2 represent continuous flooding, intermittent irrigation, and control irrigation (W2), respectively. Urea application at rates of 0 (N0), 60 (N60), 80 (N80), 100 (N100), 120 (N120), and 140 (N140) kg N ha^−1^. Values followed by the different letter are significant different at *p* < 0.05.

**Figure 7 ijerph-19-16506-f007:**
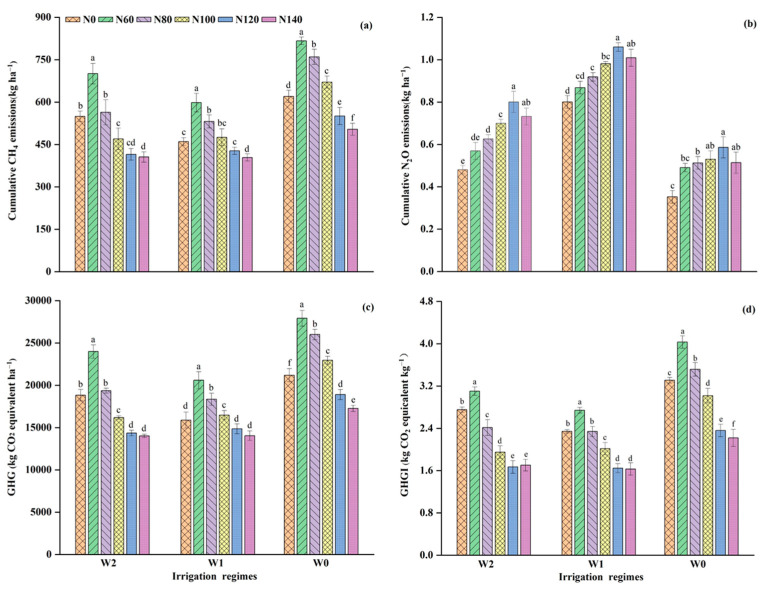
Cumulative emissions of (**a**) CH_4_, (**b**) N_2_O, (**c**) total global warming potential (GWP), and (**d**) greenhouse gas intensity (GHGI) affected by irrigation regimes with different N application rates. Different lowercase letters above the standard error bars indicate significant differences among treatments during the rice-growing season (*p* < 0.05). Values followed by the different letter are significant different at *p* < 0.05.

**Table 1 ijerph-19-16506-t001:** Effective tillering numbers, grain numbers per spike, 1000-kernel weight, and grain yield induced by water irrigation regime and N fertilization over the rice-growing season.

Water Regimes	Treatment	Effective Tillering Numbers	Grains Numbers per Panicle	1000-Kernel Weight (g)	Grain Yield (kg ha^−1^)
Flooding Irrigation (W0)	W0N0	10.7 ± 0.4 e	52.8 ± 0.2 d	28.4 ± 0.4 a	6830.0 ± 120.2 e
	W0N60	16.1 ± 0.4 cd	54.6 ± 0.3 d	29.4 ± 0.4 a	7387.6 ± 121.3 d
	W0N80	16.5 ± 0.5 c	69.1 ± 1.1 d	26.9 ± 0.6 bc	7886.7 ± 164.6 cd
	W0N100	19.5 ± 0.5 ab	85.8 ± 1.4 c	26.5 ± 0.7 bc	8114.7 ± 158.7 bc
	W0N120	20.4 ± 0.3 a	91.4 ± 0.9 a	27.5 ± 0.5 b	8550.0 ± 180.5 a
	W0N140	19.3 ± 0.3 ab	89.2 ± 0.8 b	26.0 ± 0.5 c	8300.0 ± 105.1 b
Intermittent irrigation (W2)	W1N0	14.3 ± 0.5 c	71.8 ± 1.2 d	27.1 ± 0.2 a	7233.3 ± 105.1 f
	W1N60	16.2 ± 0.6 ac	74.4 ± 1.2 d	27.2 ± 0.2 a	8013.2 ± 120.6 e
	W1N80	18.7 ± 0.8 ab	97.1 ± 1.7 b	27.8 ± 0.2 a	8370.0 ± 86.1 d
	W1N100	19.7 ± 0.8 a	101.8 ± 1.8 a	27.7 ± 0.2 a	8716.7 ± 87.8 b
	W1N120	16.6 ± 0.6 b	96.5 ± 1.2 bc	27.1 ± 0.2 a	9633.6 ± 220.4 a
	W1N140	14.3 ± 0.5 c	71.8 ± 1.2 d	27.1 ± 0.2 a	9187.0 ± 165.6 c
Controlled irrigation (W3)	W2N0	10.7 ± 0.4 e	52.8 ± 0.2 d	28.4 ± 0.4 a	6841.4 ± 25.6 e
	W2N60	9.2 ± 1.2 e	60.2 ± 0.8 e	27.2 ± 1.1 b	7732.3 ± 49.7 d
	W2N80	14.2 ± 0.5 d	69.3 ± 1.1 c	26.9 ± 0.5 c	8017.7 ± 116.5 bc
	W2N100	16.4 ± 0.6 c	73.3 ± 1.1 d	25.9 ± 0.5 c	8300.9 ± 156.4 b
	W2N120	18.6 ± 0.7 b	85.1 ± 1.2 b	25.3 ± 0.6 c	8601.7 ± 113.3 a
	W2N140	20.0 ± 0.4 a	94.8 ± 1.1 a	27.5 ± 0.6 a	8234.8 ± 186.5 b

Note: Values followed by the different letter are significant different at *p* < 0.05.

**Table 2 ijerph-19-16506-t002:** Correlation analysis between rice yield and yield components.

Trait	Effective Tillering	Grains per Panicle	1000-Kernel Weight	Yield
Effective tillerings	1			
Grains numbers per panicle	0.810 **	1		
1000-kernel weight	0.180 ns	0.286 ns	1	
Yield	0.783 **	0.927 **	0.179 ns	1

Note: “ns” means no significant difference; ** mean significant difference at the level of 0.05 and 0.01, respectively.

**Table 3 ijerph-19-16506-t003:** Two-way ANOVA analysis of rice yield; average CH_4_ and N_2_O flux; and seasonal emissions of CH_4_, N_2_O, GWP, and GHGI under N fertilization and water regimes (W), as well as their interactions (n = 3).

Source	Grain Yield	Average CH_4_ Flux	Average N_2_O Flux	CH_4_ Emissions	N_2_O Emissions	GWP	GHGI
W	**	**	ns	**	ns	**	*
N	**	ns	**	*	**	ns	**
W × N	ns	*	*	**	*	**	**

ns: insignificant; *: significant at 0.05 level; **: significant at 0.01 level. GWP and GHGI represent the global warming potential and total GWP in terms of grain yield, respectively. ANOVA: analysis of variance.

## Data Availability

The data presented in this study are available on request from the corresponding author.

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
