# Peer review of "Combination of Water-Saving Irrigation and Nitrogen Fertilization Regulates Greenhouse Gas Emissions and Increases Rice Yields in High-Cold Regions, Northeast China"

_ijerph, 2022, doi:10.3390/ijerph192416506_

Round 1
Reviewer 1 Report
This study determined the efficient water regime-N fertilizer incorporation to minimize the impact of rice production on GHG emissions in highly-cold rice-cropping system. As this is a well written paper with strong analysis of datasets, I believe the manuscript is acceptable for publication after moderate changes/corrections. I would recommend that the authors comb through the paper to check for the usage of wrong words and grammar. My comments are primarily editorial with a few more substantive, see below.
1. The objectives of the study were clear, yet results are not clear. It is also important that the authors stress the significant points of the manuscript.
2. More quantitative data are heavily needed to present the results in the abstract.
3. The figures are not concise enough. Most of them are histograms, and bars including the lowercase letters in the figures are too compact. Please check and improve all the figures and tables to make it clearer, such as using different colors, or line types.
4. The height of the gas chambers is only 50 cm, yet the height of the rice plant is about 100 cm. How are the rice plants put into t chambers as the gas is collected?
5. From the figure of greenhouse gas emission, the result of methane emission is too simple to describe, which should be further clarified.
6. Avoid using abbreviations and acronyms in the conclusions section. Remember that the conclusions must be self-explanatory.
7. It is necessary for the authors to review carefully the manuscript. It is recommended that the authors should pay attention to the use of short sentences as much as possible throughout the manuscript to accurately describe the actual meaning of each sentence. It is easy to cause logical discontinuities and confusion.
8. The study about methane emissions in rice fields is critically important. The authors found that methane emissions are significantly affected by nitrogen fertilizer and water. Whether can the application of combination of water-saving irrigation and nitrogen fertilizer be really applied in rice production practice?
Author Response
This study determined the efficient water regime-N fertilizer incorporation to minimize the impact of rice production on GHG emissions in highly-cold rice-cropping system. As this is a well written paper with strong analysis of datasets, I believe the manuscript is acceptable for publication after moderate changes/corrections. I would recommend that the authors comb through the paper to check for the usage of wrong words and grammar. My comments are primarily editorial with a few more substantive, see below.
- The objectives of the study were clear, yet results are not clear. It is also important that the authors stress the significant points of the manuscript.
Reply: We agreed to your comments about the unclear results in this study. The “Results” were further improved to make it clearer to be understood. And we emphasized that intermittent irrigation in combination with 120 kg N ha−1 N application can reduce greenhouse gas emissions t and achieve the safe production of rice in cold areas in highly-cold regions of northern China .
- More quantitative data are heavily needed to present the results in the abstract.
Reply: We thank the editor for the helpful suggestions. Some quantitative data have added in the revised manuscript.
- The figures are not concise enough. Most of them are histograms, and bars including the lowercase letters in the figures are too compact. Please check and improve all the figures and tables to make it clearer, such as using different colors, or line types.
Reply: Thanks. All the figures and tables were re-plotted to improve the visibility and clarity.
- The height of the gas chambers is only 50 cm, yet the height of the rice plant is about 100 cm. How are the rice plants put into t chambers as the gas is collected?
Reply: Thanks for catching this. When the height of the plant is more than 50cm, a sampling box was added when sampling. The height of the two boxes is 100cm.
- From the figure of greenhouse gas emission, the result of methane emission is too simple to describe, which should be further clarified.
Reply: Thanks. In the following work, we monitored the greenhouse gas emission for more than two years and explored the changes with the soil and environmental factors as much as possible and elucidate the mechanism of emissions.
- Avoid using abbreviations and acronyms in the conclusions section. Remember that the conclusions must be self-explanatory.
Reply: Thank you for your suggestions. We have replaced the abbreviations and acronyms with the terms in the revised manuscript. And we have tried our best to make the conclusions to be clear and self-explanatory.
- It is necessary for the authors to review carefully the manuscript. It is recommended that the authors should pay attention to the use of short sentences as much as possible throughout the manuscript to accurately describe the actual meaning of each sentence. It is easy to cause logical discontinuities and confusion.
Reply: Thanks, we agree with the comment. We carefully checked and revised the long sentences throughout the manuscript. The verbose and obscure sentences were improved as much as possible to make them to be easily understood.
- The study about methane emissions in rice fields is critically important. The authors found that methane emissions are significantly affected by nitrogen fertilizer and water. Whether can the application of combination of water-saving irrigation and nitrogen fertilizer be really applied in rice production practice?
Reply: This paper indicated that combination of water-saving irrigation and nitrogen fertilizer regulated greenhouse gas emissions and increases rice yields in high-cold Harbin City, which holds great potential in rice-planting and worth being popularized in Heilongjiang Province , highly-cold belt of China.
7. It is necessary for the authors to review carefully the manuscript. It is recommended that the authors should pay attention to the use of short sentences as much as possible throughout the manuscript to accurately describe the actual meaning of each sentence. It is easy to cause logical discontinuities and confusion.
Reply: Thanks, we agree with the comment. We carefully checked and revised the long sentences throughout the manuscript. The verbose and obscure sentences were improved as much as possible to make them to be easily understood.
8.The study about methane emissions in rice fields is critically important. The authors found that methane emissions are significantly affected by nitrogen fertilizer and water. Whether can the application of combination of water-saving irrigation and nitrogen fertilizer be really applied in rice production practice?
Reply :This paper indicated that combination of water-saving irrigation and nitrogen fertilizer regulated greenhouse gas emissions and increases rice yields in high-cold Harbin City, which holds great potential in rice-planting and worth being popularized in Heilongjiang Province , highly-cold belt of China.

Reviewer 2 Report
The authors presented a combination of water-saving irrigation and nitrogen fertilization regulates greenhouse gas emissions and increases rice yields in high cold regions. The data obtained is interesting. The manuscript requires a minor correction. Detailed comments:
1. Please explain all abbreviations in the abstract and check in the article (eg GWP).
2. Materials and methods - It is worth considering adding a map with the location of the research site.
3. Why did the authors add in Figure 1 (line 138-139 and 222-223) twice?
4. Figures 2, 3, 4, 5 and 6. Please enlarge the figures, axis descriptions and the legend, as it is currently not very legible.
5. It is important to check that the writing text clearly expresses and explains each idea and result obtained.
6. Please add DOI links in your reference list.
Author Response
The authors presented a combination of water-saving irrigation and nitrogen fertilization regulates greenhouse gas emissions and increases rice yields in high cold regions. The data obtained is interesting. The manuscript requires a minor correction. Detailed comments:
1.Please explain all abbreviations in the abstract and check in the article (eg GWP).
Reply:Thanks. We have checked all abbreviations throughout this manuscript.
2. Materials and methods - It is worth considering adding a map with the location of the research site.
Reply: Thanks for the valuable suggestions. We have added a map of the research site (Figure 1) in the revised manuscript
3. Why did the authors add in Figure 1 (line 138-139 and 222-223) twice?
Reply: We are very sorry for our carelessness. We have deleted the repeated the sentences in line 138-139 in the revised manuscript.
4. Figures 2, 3, 4, 5 and 6. Please enlarge the figures, axis descriptions and the legend, as it is currently not very legible.
Reply: Thanks. Made the changes as suggested.
5. It is important to check that the writing text clearly expresses and explains each idea and result obtained.
Reply: Made the changes as suggested.
6. Please add DOI links in your reference list.
Reply: Thanks. Made the changes as suggested.